# Colchicine-Induced Polyploidy in *Rhododendron fortunei* Lindl

**DOI:** 10.3390/plants9040424

**Published:** 2020-03-31

**Authors:** Lan Mo, Junhao Chen, Xiongzhen Lou, Qiangwei Xu, Renhui Dong, Zaikang Tong, Huahong Huang, Erpei Lin

**Affiliations:** State Key Laboratory of Subtropical Silviculture, Zhejiang A&F University, Hangzhou 311300, Zhejiang, China; molan9054@gmail.com (L.M.); hanschen001@gmail.com (J.C.); xzlou@zafu.edu.cn (X.L.); xingyaoqiang009@gmail.com (Q.X.); 20143025@zafu.edu.cn (R.D.); zktong@zafu.edu.cn (Z.T.); huanghh@zafu.edu.cn (H.H.)

**Keywords:** polyploid, tetraploidy, plant stomata, flow cytometry

## Abstract

Polyploidy in *Rhododendron fortunei* has great potential to improve its horticultural and commercial value, and to also meet market demands. In this study, a feasible method for polyploid induction in *R. fortunei* via colchicine treatment was established, and the obtained polyploid plants were identified and characterized. As a result, the stem bases of tissue-cultured plantlets treated with 0.1% colchicine for 24 h showed the highest polyploid induction with a rate of 36.67%. By flow cytometric analysis, 69 tetraploids and 29 octoploids were identified in the regenerated plants that were examined. Phenotypic analysis indicated that the leaves of tetraploid and octoploid plants were smaller, rounder and thicker with more abundant and longer epidermal hairs than those of diploids. Furthermore, the stomata of polyploids were larger and sparser than those of diploids. An increase in chlorophyll content was also detected in polyploids, which resulted in darker green leaves. In conclusion, our study established an effective method to induce polyploidy in *R. fortunei*, which could be used to develop new genetic resources for breeding *R. fortunei* and other *Rhododendron* species in the future.

## 1. Introduction

*Rhododendron fortunei* Lindl, one of the alpine rhododendrons, is an evergreen small tree or shrub of genus *Rhododendron*, and mainly grows at an altitude of 620 to 2000 meters in Southern China [1]. Owing to the beautiful and bright color of its flowers, and their elegant shape and charming aroma, *R. fortunei* has a high ornamental value and great commercial potential in landscaping and horticultural markets [2]. However, the use of *R. fortunei* in landscaping and horticultural markets is severely limited by its slow growth rate and reliance on an alpine climate. Therefore, its environmental adaption and horticultural traits need to be improved in order to advance the development and utilization of *R. fortunei*.

Naturally occurring polyploidy is widespread in the plant kingdom, and polyploidization is often responsible for the generation of new species during plant evolution [3]. Compared with their diploid relatives, polyploid plants often exhibit advantageous features, including greater adaptability and resistance, and they have giant flowers, fruits, and other prominent features [4,5,6]. To exploit these advantages, artificial methods of inducing polyploidy have become an important tool for plant breeding and they have been widely applied in the breeding programs of crops, vegetables and flowers [7]. Herbicides, such as colchicine, trifluralin and oryzalin, are chemicals that are frequently used to induce polyploidy in plants, as they effectively arrest mitosis at the anaphase stage [8,9]. Polyploids of several *Rhododendron* species have been successfully induced using these herbicides, resulting in varieties with improved ornamental value that provide new genetic resources for crossbreeding [10,11,12]. However, at present, there are no reported instances of polyploidy induction in *R. fortunei* species. Therefore, it is important to establish a polyploidy induction method for generating new breeding resources, which should help to improve the horticultural traits of *R. fortunei*.

The aim of our study was to establish a highly efficient method for polyploidy induction in *R. fortunei*. The shoot apex and stem bases of aseptic tissue-cultured plantlets were used as explants, and colchicine treatments of varying concentration and time were assessed. The ploidy level of regenerated plants was estimated by flow cytometry, and the morphological and physiological characteristics of polyploids were also determined.

## 2. Results

### 2.1. Effects of Colchicine Treatment on Explants’ Survival

The tissue-cultured plantlets of *R. fortunei* clone ZL1 were used as explants for polyploidy induction. After colchicine treatment, the treated explants were incubated in regeneration medium for 15 days, and the survival of explants was assessed by the mortality rate. Basically, the mortality rate of explants increased with an increase in colchicine concentration and treatment time (Figure 1). Colchicine treatment for 24 h showed the least toxic effect with a 3.3%–31.3% mortality rate for shoot apices and stem bases, respectively, whereas 72 h of treatment showed the strongest toxic effect with a mortality rate up to 80% (Figure 1B). Furthermore, the mortality rate of shoot apices was much lower than that of stem bases, and stem bases had a greater than 50% mortality rate when treated for 72 h with different colchicine concentrations (Figure 1B).

### 2.2. Polyploid Identification by Flow Cytometry

Ploidy levels of the regenerated plantlets from various colchicine treatments were estimated by flow cytometry and FCS Express software. Thirty plantlets from each treatment group were chosen for detection. As shown in Figure 2 and Table 1, four different patterns of flow cytometry were observed. The uninduced diploid plantlets showed a large peak at 25.26 on average (Figure 2A, Table 1), whereas some of the induced plantlets showed a large peak shift to 48.92 or 94.94 on average, which represent tetraploids and octoploids, respectively (Figure 2B,C, Table 1). Interestingly, some plants had two peaks at around 25.25 and 52.51, indicating that they were chimeras (Figure 2D, Table 1). In total, 69 tetraploids and 29 octoploids were obtained from 540 regenerated plantlets from all of the examined colchicine treatments (Table 1 and Table 2).

The number of polyploids resulting from different colchicine treatments were also compared (Table 2). The outcome suggests that of all the treatments, treatment with 0.1% colchicine for 24 h showed the highest induction rate of 33.33% (20/60), whereby 12 tetraploids, four octoploids, and four chimeras were obtained (Table 2). In comparison, the induction rates for the other treatments were much lower. Furthermore, colchicine treatment on stem bases exhibited a higher polyploid induction rate. When treated with 0.1% colchicine for 24 h, the polyploid induction rate of stem bases reached up to 36.67% with 26.67% (8/30) tetraploids and 10% (3/30) octoploids, respectively (Table 2). By contrast, the tetraploid and octoploid induction rates of shoot apices were only 13.33% (4/30) and 3.33% (1/30), respectively (Table 2). This outcome suggests that the stem bases are more sensitive to colchicine treatment and are a better choice of explant for polyploid induction. Considering its low mortality rates (15.83%), treatment of 0.1% colchicine on stem bases for 24 h was considered as the optimal condition for polyploid induction in tissue-cultured plantlets of *R. fortunei*.

### 2.3. Morphological Characteristics of R. fortunei Polyploids

After the ploidy levels were analyzed, plantlets with different ploidy levels were transplanted to a greenhouse and grown for a further six months. The morphological characteristics of these six-months old plants were investigated and diploid and polyploid plants were compared. As shown in Figure 3, the leaves of tetraploids and octoploids were considerably rounder, harder and thicker with a slightly rough surface and darker color compared to those of the diploids. For the diploid plants, the mean leaf length was 4.34 cm with a mean leaf width of 3.20 cm and mean leaf area of 10.81 cm^2^ (Table 3). For tetraploid and octoploid plants, the mean leaf length was 2.34 cm and 1.37 cm, the mean leaf width was 2.04 cm and 1.29 cm, and the mean leaf area was 3.74 cm^2^ and 1.37 cm^2^, respectively (Table 3). This showed that that the polyploid leaves were dramatically smaller than the diploid leaves. In addition, the leaves of diploid plants showed an elliptical shape with a mean leaf index of 1.35, whereas the leaves of tetraploids and octoploids were much rounder with a mean leaf index of 1.14 and 1.06, respectively, which are very close to 1.00 (Table 3). Additionally, the thickness of leaves increased with ploidy level, the mean thickness of octoploids leaves was 0.753 mm, which was much thicker than that of tetraploids (0.574 mm) and diploids (0.338 mm) (Table 3). Leaf epidermal hairs were also observed under the stereoscope. As shown in Figure 4, denser and longer epidermal hairs were observed on leaf surfaces and edges of octoploids and tetraploids, which meant they had a rougher leaf surface than diploids. 

Furthermore, to evaluate the stability in the morphology of these polyploid plants, the leaf characteristics of 15-months old plants were also investigated and compared. As a result, significant differences were also observed between diploids and polyploids (Figure 5). Compared with diploids, the polyploid plants still exhibited rounder, smaller and thicker leaves (Figure 5, Table 3). The mean leaf area of diploid plants was 26.91 cm^2^, while that of tetraploids and octoploids were only 17.05 cm^2^ and 3.72 cm^2^, respectively (Table 3). The average leaf length and width of diploid plants were 9.63 cm and 3.55 cm, while that of tetraploids and octoploids were 6.57 cm and 3.26 cm, and 2.43 cm and 1.75 cm, respectively (Table 3). The average leaf index value also indicated that the leaves of tetraploids (mean leaf index =2.04) and octoploids (mean leaf index =1.39) became much rounder than those of diploids (mean leaf index =2.71) (Table 3). In particular, the growth of octoploids were severely disrupted with extremely small and deformed leaves (Figure 5, Table 3), which means octoploids maybe not be a suitable genetic resource for cross breeding *R. fortunei*. These results indicate that increased ploidy level causes obvious morphological changes in polyploids of *R. fortunei*. These morphological characteristics are easily identifiable and may be used to preliminarily identify putative *R. fortunei* polyploids.

### 2.4. Stomatal Characteristics of Polyploid R. fortunei 

In general, polyploids have larger stomata at lower density compared to those of diploids, which is one of the hallmark characteristics of polyploidy [13,14]. In our study, stomatal size and density were also compared between diploids and polyploids. There were significant differences observed in stomatal size and density among diploids, tetraploids and octoploids (Figure 6). The mean stomata length and width of diploid plants was 27.30 μm and 21.19 μm, respectively, while the mean stomata length and width of tetraploids and octoploids increased to 34.07 μm and 30.41 μm, and 42.67 μm and 38.89 μm, respectively (Table 4). The average stomatal area of diploids (454.02 μm^2^) was smaller than that of tetraploids and octoploids (814.77 μm^2^ and 1307.44 μm^2^ on average, respectively; Table 4). Moreover, the stomatal density of polyploids was significantly lower than that of diploid plants (Figure 6). On average, there were 164.79 stomata/mm^2^ in diploid plants, whereas an average of 78.93 and 52.40 stomata/mm^2^ were observed in tetraploids and octoploids, respectively (Table 4). 

### 2.5. Pigment Content in Leaves of Polyploid and Diploid R. fortunei

From the morphological observations, it was obvious that polyploid leaves had a darker green hue than diploid leaves. The estimation of pigment content revealed that tetraploid and octoploid leaves had significantly higher chlorophyll content than that of diploids (Table 5). On average, 76.42 mg/g chlorophyll a, 41.38 mg/g chlorophyll b and 117.80 mg/g total chlorophyll, and 48.57 mg/g chlorophyll a, 27.80 mg/g chlorophyll b and 76.37 mg/g total chlorophyll were detected in octoploids and tetraploids, respectively, whereas only 28.55 mg/g chlorophyll a, 12. 64 mg/g chlorophyll b and 41.19 mg/g total chlorophyll were detected in diploid plants (Table 5). In contrast, the carotenoid content in diploids was much higher than that of polyploid plants. As shown in Table 5, the mean carotenoid content of diploids was 17.83 mg/g, while the mean content for tetraploids and octoploids was 14.71 mg/g and 14. 70 mg/g, respectively.

## 3. Discussion

In the agricultural and horticultural industries, polyploid cultivars are usually of higher value as they often possess superior agronomic traits compared to their diploid counterparts. Artificial polyploid induction, which represents a useful method for breeders, has been widely applied in the breeding programs of crops, flowers, and economically important trees [15]. However, *R. fortunei* polyploid induction by colchicine has not previously been reported. The ability to induce polyploidy in this species is important to improve its horticultural traits and commercial value.

Colchicine is a chemical mutagen frequently used to induce polyploidy, and there are many instances whereby colchicine has been used successfully to induce polyploidy in multiple plants [16,17,18]. These studies suggest that colchicine concentration and treatment time are major factors that influence polyploid induction. In our study, the effects of colchicine treatment concentration and time were first assessed based on the mortality rates and polyploid induction rates. Generally, the mortality rate of *R. fortunei* explants increased with colchicine concentration and treatment time (Figure 1). The toxic effect of high doses of colchicine has also been reported in other polyploid induction studies [19]. Conversely, the polyploid induction rate did not increase with colchicine concentration and time (Figure 1). Based on flow cytometric analysis of regenerated plantlets, the highest polyploid induction rate (33.33%) was achieved by applying 0.1% colchicine for 24 h, and other treatments including higher colchicine concentrations and longer treatment time did not result in a greater proportion of polyploids (Table 2). Similarly, in a previous study on polyploid induction of three *Rhododendron* cultivars, most tetraploids were obtained by treatment with 0.05% colchicine for 48 h, which resulted in 4.8% tetraploids [20]. Thus, it is necessary to determine optimal colchicine concentrations and treatment times for polyploid induction in different plants. 

Flow cytometric analysis and chromosomal observations are conventional and reliable methods to determine ploidy level in plants. Although chromosome analysis is a more direct way to verify ploidy level, it is not suitable for *R. fortunei* and other *Ericaceae* species because they have small chromosomes that are difficult to see and count using standard cytological techniques [21]. Flow cytometry has been proven to be a rapid and efficient method for estimating ploidy level in almost all regenerated plant tissues, and it is particularly suited to studies involving large sample numbers [19,22]. Jones et al. used flow cytometry to determine the ploidy level and genome size of 200 diverse species and cultivars of *Rhododendron*, which now serves a valuable database for breeders. Flow cytometry was also used to determine the ploidy level of induced polyploid adventitious shoots of three *Rhododendron* hybrids [11,23]. In our study, polyploid *R. fortunei* plants were identified from hundreds of plantlets by flow cytometric analysis prior to morphological observations (Table 2). This method greatly improved the efficiency of polyploid isolation and reduced the labor input of plant cultivation.

Notably, colchicine treatment on stem bases as explants yielded higher polyploid induction rates than shoot apices in our study (Table 2). It is possible that removing the apex is beneficial to the regeneration of adventitious buds, which promote polyploid induction after colchicine treatment. In *Rhododendron racemosum* Franch, a polyploid induction rate of 32% was obtained from 0.15% colchicine for 24 h treatment on the stem bases of plantlets [23]. However, in hot pepper (*Capsicum annuum*), the polyploid induction rate of treatment on shoot apex was much higher than treatment on shoot bases [24]. Thus, the optimal explant type for colchicine polyploid induction is possibly different for different plant species and should be experimentally verified in each case.

Generally, increases in ploidy level result in obvious morphological changes, which are reliable indicators of polyploidy [7,25,26]. Morphological changes in leaves are more easily observed and characterized. Induced tetraploids usually have larger leaves with an obviously altered shape compared to their diploid counterparts [27,28]. Consistent with previous studies, significant phenotypic differences were observed in the leaves of *R. fortunei* diploids and polyploids. Specifically, the leaves of tetraploid and octoploid *R. fortunei* plants were considerably rounder, harder and thicker with a rougher leaf surface (Figure 3 and Figure 5, Table 3). Interestingly, leaves of polyploid *R. fortunei* were significantly smaller than that of diploid leaves (Figure 3 and Figure 5, Table 3), which is not commonly found in polyploidization. However, reduced leaf size during polyploid induction is a frequently observed phenomenon. For instance, in tillered onion (*Allium cepa L. var. Agrogatum Don*), the polyploids induced by colchicine and pendimethalin treatment had much smaller leaves and reduced plant height compared to its diploid counterpart [29]. Eeckhaut et al. obtained a tetraploid *R. simsii* that showed smaller and rounder leaves and grew more slowly than its diploid [30]. Lyuchun, P et al. also found that there were smaller and larger leaves were observed in different colchicine induced polyploids of *R. racemosum Franch* [23]. Besides, in colchicine-induced polyploid *Lilium pumilum* DC, compared with the diploid plant, the growth of tetraploid leaves was shortened and the plant grew slower in the early stage, while the tetraploid plant accelerated growth in the later stage, and the size of leaf increased obviously [31]. So, we proposed the hypothesis that accumulation of colchicine in the polyploidy plantlets influences the mitosis, which leads to the limited size of leaves. Another possible explanation for this phenomenon is the increased demand for cellular resources and energy to support cell division in polyploids [32]. It is conceivable that genome duplication results in a need for greater resources to replicate the genome before cell division, thus creating a higher demand for resources in polyploid plants [33]. A disturbance in cell division may result in smaller organ size with higher ploidy level. 

Stomata characteristics are another frequently used indicator of the ploidy level of plants [34]. Compared with their diploid counterparts, the stomata of induced *R. fortunei* polyploids were larger, whereas their stomatal density sharply decreased (Figure 6, Table 4). This is consistent with many previous studies. The stomata of tetraploid passion fruit (*Passiflora edulis Sims.*) was significantly larger than those of diploid counterparts, whereas stomatal density was significantly reduced with higher ploidy [35]. Stomata of tetraploid *Gladiolus grandiflorus* had increased length and width but decreased density [36]. According to these studies, stomata characteristics may be a more reliable indicator of polyploids [37,38].

Higher chlorophyll content is a frequently observed event in polyploid induction in diverse plant species [39,40]. In our study, higher chlorophyll content was detected in polyploid *R. fortunei* plantlets, explaining the observed darker green of polyploid leaves (Table 5). Similarly, tetraploids of cassava (*Manihot esculenta*) were found to have higher chlorophyll content, which increased photosynthetic capacity and led to higher yield compared with diploids [40]. In *Stevia rebaudiana*, tetraploids also had a significantly higher chlorophyll content than diploid plants, resulting in higher photosynthetic capacity [39]. 

In summary, it is possible to obtain induced polyploids from tissue-cultured plantlets of *R. fortunei* using colchicine treatments, and an optimal induction method was established based on systematic screening of colchicine concentration and treatment time. However, the efficiency of tetraploid induction can still be improved. As a valuable ornamental plant, polyploid induction will provide new genetic resources for future breeding programs of *R. fortunei* and *Rhododendron* species. Polyploid induction also provides the opportunity to improve the seed or pollen vigor of the subgenus hybrid between evergreen and deciduous *Rhododendron* species. The observed morphological changes resulting from polyploidization require further characterization. The obtained polyploids are currently maintained in a greenhouse for further use in breeding programs once they flower.

## 4. Materials and Methods

### 4.1. Plant Materials

*R. fortunei* clone ‘ZL1’ came from a superior tree of Huading mountain in Zhejiang province, China. This clone exhibits superior horticultural traits including a long flowering time and strong resistance and it has been successfully propagated through tissue culture. In this study, tissue-cultured plantlets of the clone ZL1 were used as the explants for polyploidy induction. These aseptic plantlets were propagated in WPM (woody plant medium) with 30 g/L sucrose and 9 g/L agar, pH5.2, and when they had grown up to about 5 cm they were used for colchicine treatment.

### 4.2. Colchicine Treatment and Shoot Regeneration

Seven colchicine concentrations (0, 0.024, 0.05, 0.1, 0.15, 0.2 and 0.25%) and three treatment times (24, 48 and 72 h) were set up, thus creating 21 experimental treatment groups. Ninety tissue-cultured plantlets were used in each treatment. All plantlets were soaked with colchicine solution for 24, 48 and 72 h, and were then thoroughly washed three times with sterilized water. Finally, the shoot apex and stem bases were cut from plantlets and transferred to regeneration medium (WPM medium with 2 mg/L ZT, 0.1 mg/L NAA, 30 g/L sucrose and 9 g/L agar, pH 5.2), respectively. Shoot regeneration was performed in a growth room at 25 ± 2 °C with 1500–2000 Lx light intensity and a 12-h light/12-h dark photoperiod [41]. After approximately two months, callus was induced from explants, and the adventitious shoots regenerated from the callus were used for further analysis.

### 4.3. Mortality Rate Determination

The mortality rate of explants was determined after 15 days of cultivation in the regeneration medium. The mortality rate was calculated as follows:Mortality rate (%) = (number of dead explants/number of treated explants) × 100.

### 4.4. Flow Cytometry for Ploidy Level Estimation

After cultivation in the regeneration medium for two months, and when the regenerated adventitious shoots grew to 1.5 cm, they were transplanted to the rooting medium (WPM medium containing 1 mg/L NAA, 30 g/L sucrose and 9 g/L agar, pH 5.2) to grow for another one month. When plantlets were approximately 3–5 cm in height, newly formed young leaves of 30 plantlets randomly chosen from each treatment were used for ploidy level estimation by flow cytometry. In total, 540 plantlets regenerated from colchicine treatments were tested by flow cytometry. Briefly, new leaves of each sample were finely chopped with a razor blade in a petri dish with 500 μL of Cystain Ultraviolet Precise P Nuclei Extraction Buffer (Sysmex Partec, Germany). The solution was filtered using CellTrics 50 μm disposable filters (Sysmex Partec, Germany) to remove leaf tissue debris. Then, the extracted nuclei were stained with 1.5 mL 4,6-diamidino-2 phenylindole (DAPI) staining buffer (Sysmex Partec, Germany), and stood at rest, avoiding light for 5–10 mins. The suspension containing stained nuclei was analyzed using CyFlow Counter flow cytometry (Sysmex Partec, Germany). When the number of nuclei was greater than 5000 and the variation coefficient (CV) value was less than 5%, data were recorded and visualized by FCS Express software. The test was repeated at least three times for each sample and the mean peak value of each sample was first calculated based on the three repeats, and then averaged for diploids, tetraploids, octoploids and chimeras, respectively. Ploidy level was determined by comparing mean peak value of each sample with that of the diploid [22]. The induction rate was calculated as follows: Induction rate (%) = number of polyploid plantlets/number of analyzed plantlets × 100.

### 4.5. Leaf Morphology Measurement

After the flow cytometric analysis, identified diploid, tetraploid and octoploid plantlets were transferred to a greenhouse, and after growing for 6 months and 15 months, the mature leaves from the middle part of the plants were sampled for morphology measurements. The leaves were imaged by a scanner, then leaf length, width, and area were determined using the Image J software and millimeter graph paper method [42]. Leaf thickness was measured using a leaf thickness gauge (YH-1, China). At least ten plants of each ploidy level were included in the leaf morphology measurement [43]. 

### 4.6. Determination of Leaf Pigment Content

Ten plants of each ploidy level were used for measurement of the pigment content in leaves. Mature leaves (0.1 g) were chopped into small pieces and soaked in 95% ethanol for 48 h in the dark [44]. Absorbance measurements of each extraction were made using a spectrophotometer at 665 nm, 649 nm, and 470 nm. The test for each sample was repeated three times, and the mean absorbance value of each sample was first calculated based on the three repeats, and then averaged for diploids, tetraploids and octoploids. According to the respective formulas for pigment concentration in alcohol extract, the contents of chlorophyll a and b, total chlorophyll and carotenoids were calculated [45]; chlorophyll content was measured by using the method of Makeen et al [46].

### 4.7. Determination of Stomatal Cell Size and Density

The abaxial leaf surface of mature leaves was coated with transparent nail polish, then the lower epidermis was peeled off and put on a glass microscope slide for stomata analyses [47,48]. Stomatal features were examined using a DXM 1200 microscope (Nikon, Japan). Stomatal length, width, and density were measured and compared between diploid and polyploid plants [49]. Stomatal density was determined by counting the number of stomata that were evenly distributed across six microscopic fields [50]. The length, width and area of 30 stomata of each ploidy level were measured and calculated.

### 4.8. Data Processing

One-way ANOVA and Duncan’s test were performed using SPSS ver.17.0 (SPSS, Chicago, IL, USA). Differences were considered statistically significant when *p* < 0.05.

## 5. Conclusion

In our study, an efficient polyploid induction method was established for tissue-cultured plantlets of *R. fortunei*. Based on mortality rate and polyploid induction rate, the optimal treatment to induce polyploids was 0.1% colchicine for 24 h. Stem bases of plantlets were shown to be more amenable to polyploid induction by colchicine treatment than shoot apices. Through ploidy level analysis, 69 tetraploids and 29 octoploids plants of *R. fortunei* were obtained from the 540 plantlets examined from all treatment groups. In comparison to diploids, polyploid plants exhibited dramatic changes in leaf characteristics, specifically smaller, rounder and thicker leaves with more and longer epidermal hairs. On the other hand, polyploids exhibited a larger stomatal apparatus, lower stomatal density and higher chlorophyll content. These changes were more apparent in octoploids than in tetraploids. These results indicate that *R. fortunei* polyploidy is successfully induced by colchicine treatment and presents an efficient and reliable methodology for breeding programs of *Rhododendron* species in the future.

## Figures and Tables

**Figure 1 plants-09-00424-f001:**
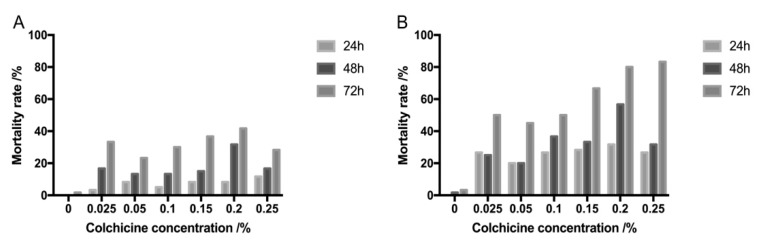
Effects of colchicine treatment on survival of *R. fortunei* explants. (**A**) Mortality rate of shoot apices; (**B**) Mortality rate of stem bases.

**Figure 2 plants-09-00424-f002:**
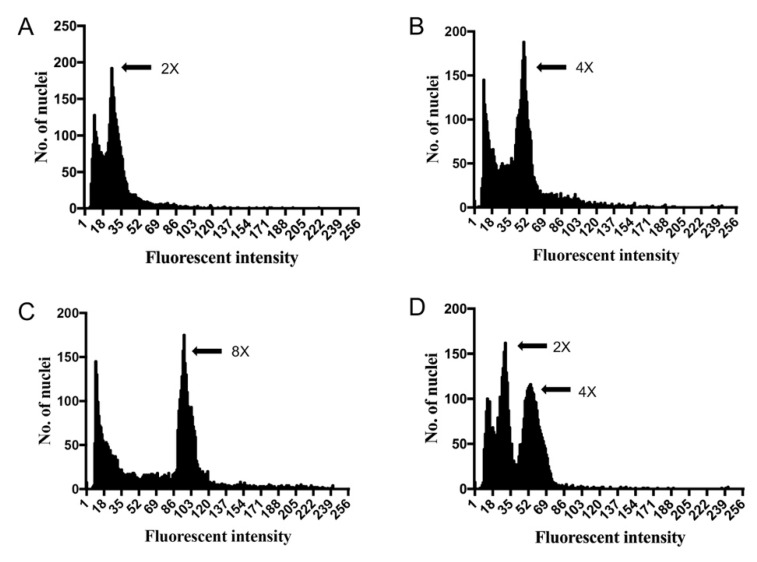
Analysis of ploidy level for induced plants by flow cytometry. (**A**) Diploid plantlets as control peak value = 25.26 ± 1.08; (**B**) Induced tetraploids, peak value = 48.92 ± 1.81; (**C**) Induced octoploids, peak value = 94.94 ± 2.44; (**D**),Induced chimera plantlets, peak value = 25.25 ± 1.33 and 52.51 ± 2.01, respectively.

**Figure 3 plants-09-00424-f003:**
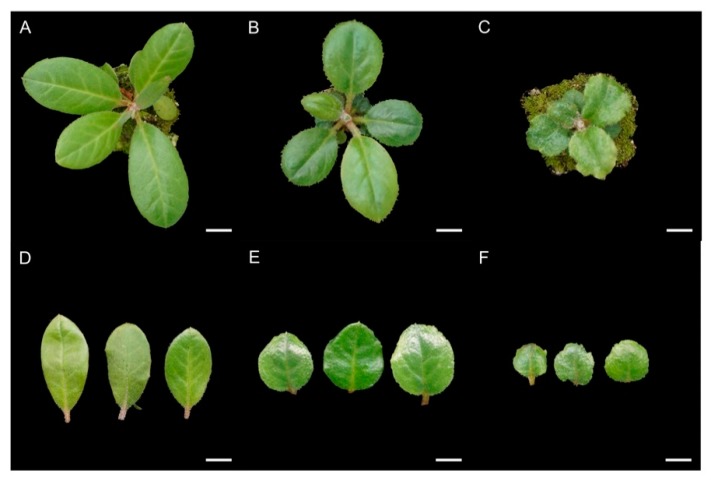
Comparison of morphological difference between 6-months old polyploid and diploid *R. fortunei* plants. (**A** and **D**), a diploid plant and its leaves; (**B** and **E**), a tetraploid plant and its leaves; (**C** and **F**), an octoploid plant and its leaves. Scale bar = 1 cm.

**Figure 4 plants-09-00424-f004:**
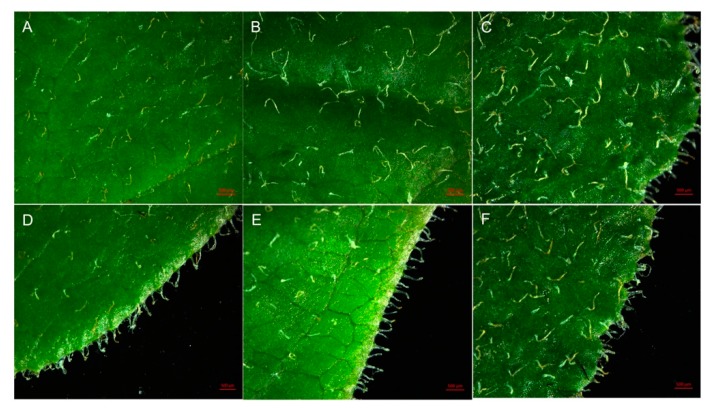
Leaf epidermal hairs of polyploid and diploid plants. **A** and **D**, epidermal hairs in the center (**A**) and on the edge (**D**) of a diploid leaf; **B** and **E**, epidermal hairs in the center (**B**) and on the edge (**E**) of a tetraploid leaf; **C** and **F**, epidermal hairs in the center (**C**) and on the edge (**F**) of an octoploid leaf. Scale bar = 500 μm.

**Figure 5 plants-09-00424-f005:**
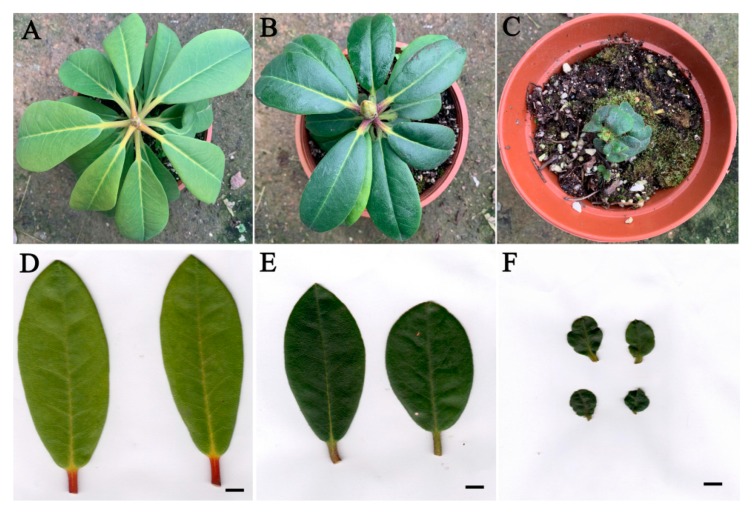
Comparison of morphological difference between 15-months old polyploid and diploid *R. fortunei* plants. (**A** and **D**), a diploid plant and its leaves; (**B** and **E**), a tetraploid plant and its leaves; (**C** and **F**), an octoploid plant and its leaves. Scale bar = 1 cm.

**Figure 6 plants-09-00424-f006:**
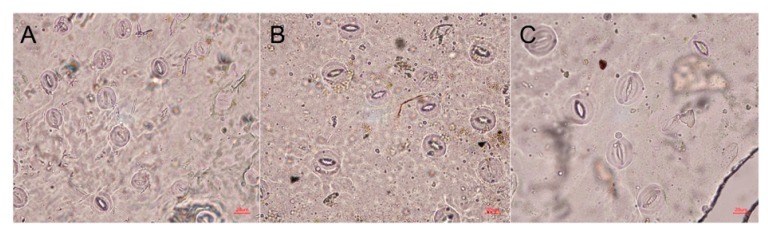
Stomata of diploid, tetraploid, and octoploid plants of *R. fortunei*. (**A**) Stomata of a diploid plant. (**B**) Stomata of a tetraploid plant. (**C**) Stomata of an octoploid plant. Scale bar = 20 μm.

**Table 1 plants-09-00424-t001:** Ploidy level analysis for induced polyploids of *R. fortunei.*

Ploidy Level	No. of Evaluated Plants	Peak Value
Diploid	405	25.26 ± 1.08
Tetraploid	69	48.92 ± 1.81
Octoploid	29	94.94 ± 2.44
Chimera	37	25.25 ± 1.33, 52.51 ± 2.01 ^a^

^a^ These two mean values represented two main peaks detected in chimera plantlets. The test for each sample was repeated three times and mean peak value of each sample was firstly calculated based on the three repeats, and then averaged for diploids, tetraploids, octoploids and chimeras, respectively.

**Table 2 plants-09-00424-t002:** Comparison of the number of polyploid induction from different colchicine treatments.

Colchicine Concentration /%	Treatment Time /h	Tetraploids Frequency		Octoploids Frequency		Chimeras Frequency		Polyploids Frequency	
		Shoot Apex	Stem Base	Shoot Apex	Stem Base	Shoot Apex	Stem Base	Shoot Apex	Stem Base
0	24	0/30	0/30	0/30	0/30	0/30	0/30	0/30	0/30
	48	0/30	0/30	0/30	0/30	0/30	0/30	0/30	0/30
	72	0/30	0/30	0/30	0/30	0/30	0/30	0/30	0/30
0.025	24	0/30	0/30	0/30	0/30	0/30	0/30	0/30	0/30
	48	0/30	0/30	0/30	0/30	0/30	0/30	0/30	0/30
	72	0/30	0/30	0/30	0/30	0/30	0/30	0/30	0/30
0.05	24	1/30	3/30	0/30	0/30	0/30	0/30	1/30	3/30
	48	1/30	3/30	0/30	2/30	1/30	2/30	2/30	7/30
	72	1/30	4/30	1/30	2/30	1/30	1/30	3/30	7/30
0.1	24	4/30	8/30	1/30	3/30	2/30	2/30	7/30	13/30
	48	1/30	4/30	0/30	1/30	2/30	1/30	3/30	6/30
	72	2/30	3/30	1/30	2/30	1/30	2/30	4/30	7/30
0.15	24	1/30	3/30	1/30	3/30	0/30	0/30	2/30	6/30
	48	1/30	4/30	0/30	2/30	1/30	3/30	2/30	9/30
	72	2/30	2/30	0/30	1/30	1/30	2/30	3/30	5/30
0.2	24	1/30	2/30	0/30	1/30	2/30	3/30	3/30	6/30
	48	2/30	3/30	1/30	1/30	1/30	2/30	4/30	6/30
	72	2/30	2/30	0/30	0/30	1/30	1/30	3/30	3/30
0.25	24	1/30	3/30	0/30	2/30	0/30	0/30	1/30	5/30
	48	1/30	2/30	0/30	1/30	2/30	1/30	3/30	4/30
	72	1/30	1/30	1/30	2/30	1/30	1/30	3/30	4/30
Total		22/540	47/540	6/540	23/540	16/540	21/540	44/540	91/540

**Table 3 plants-09-00424-t003:** Leaf morphological characteristics of polyploid and diploid *R. fortunei* plants.

Plant Age	Ploidy Level	No. of Evaluated Plants	Leaf Length (cm)	Leaf Width (cm)	Leaf Index	Leaf Area (cm^2^)	Thickness (mm)
6 months	Diploid	10	4.34 ± 1.00 ^a^	3.20± 0.51 ^a^	1.35 ± 0.32 ^a^	10.81 ± 5.10 ^a^	0.338 ± 0.027 ^c^
	Tetraploid	10	2.34 ± 0.50 ^b^	2.04 ±0.22 ^b^	1.14 ± 0.20 ^b^	3.74 ± 1.22 ^b^	0.547 ± 0.021 ^b^
	Octoploid	10	1.37 ± 0.47 ^c^	1.29 ± 0.35 ^c^	1.06 ± 0.14 ^b^	1.37 ± 1.35 ^c^	0.753 ±0.031 ^a^
15 months	Diploid	10	9.63 ± 0.78 ^a^	3.55 ± 0.39 ^a^	2.71 ± 0.27 ^a^	26.91 ± 4.03 ^a^	0.458 ± 0.047 ^c^
	Tetraploid	10	6.57 ± 1.03 ^b^	3.26 ±0.35 ^b^	2.04 ± 0.30 ^b^	17.05 ± 5.12 ^b^	0.675 ± 0.043 ^b^
	Octoploid	10	2.43 ± 0.30 ^c^	1.75 ± 0.24 ^c^	1.39 ± 0.11 ^c^	3.72 ± 0.41 ^c^	0.905 ±0.053 ^a^

Ten plants of each ploidy level were included in the morphology measurement, and mean values ± SE in the same column followed by different lower letter are significantly different at *p* < 0.05 according to Duncan’s test.

**Table 4 plants-09-00424-t004:** Stomatal characteristics of polyploid and diploid *R. fortunei* plants.

Ploidy Level	No of Evaluated Stomata	Stomata Length (μm)	Stomata Width (μm)	Stomata Index (Length/Width)	Stoma Area (μm^2^)	Stoma Density (mm^−2^)
Diploid	30	27.30 ± 0.66 ^c^	21.19 ± 1.18 ^c^	1.30 ± 0.07 ^a^	454.02 ± 27.83 ^c^	164.79±11.85 ^a^
Tetraploid	30	34.07 ± 1.93 ^b^	30.41 ±1.87 ^b^	1.12 ± 0.06 ^b^	814.77 ± 82.39 ^b^	78.93±7.72 ^b^
Octoploid	30	42.67 ± 2.99 ^a^	38.89 ± 3.00 ^a^	1.10 ± 0.07 ^b^	1307.44 ± 176.26 ^a^	52.40±5.38 ^c^

Thirty stomata were measured for each ploidy level, and mean values ± SE in the same column followed by different lower letter are significantly different at *p* < 0.05 according to Duncan’s test.

**Table 5 plants-09-00424-t005:** Content of leaf pigments in diploid, tetraploid, and octoploid *R. fortunei.*

Ploidy Level	No. of Evaluated Plants	Total Chlorophyll (mg/g)	Chlorophyll A (mg/g)	Chlorophyll B (mg/g)	Carotenoid (mg/g)
Diploid	10	41.19 ± 1.29 ^c^	28.55 ± 1.89 ^c^	12.64 ± 2.33 ^c^	17.83 ± 1.47 ^a^
Tetraploid	10	76.37 ± 3.58 ^b^	48.57 ± 2.15 ^b^	27.80 ± 2.47 ^b^	14.71 ± 0.73 ^b^
Octoploid	10	117.80 ± 6.63 ^a^	76.42 ± 4.50 ^a^	41.38 ± 3.57 ^a^	14.70 ± 2.12 ^b^

Ten plants of each ploidy level were used for measurement of pigment content in leaves. The test for each sample was repeated three times, and mean absorbance value of each sample was firstly calculated based on the three repeats, and then averaged for diploids, tetraploids and octoploids, respectively. Mean values ± SE in the same column followed by different lower letter are significantly different at *p* < 0.05 according to Duncan’s test.

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
