# Peer review of "Colchicine-Induced Polyploidy in Rhododendron fortunei Lindl"

_plants, 2020, doi:10.3390/plants9040424_

Round 1

Reviewer 1 Report

Report on manuscript

ID plants-732395

Colchicine-induced Polyploidy in Rhododendron fortunei Lindl.

Authors

Lan Mo, Junhao Chen, Qiangwei Xu, Xiongzhen Lou, Renhui Dong, Huahong Huang, Zaikang Tong, Erpei Lin

The study is principally descriptive, the aim of the authors was to induce polyploidy in diploids of Rhododendron fortunei Lindl., to evaluate methodology for polyploid induction and to describe newly arisen polyploids. These aims were fully met.

The study has brought new and original data on polyploid induction in Rhododendron. I feel that the topic is not of general interest, but is of high value for specialist in polyploidy research and Rhododendron breeders. Polyploidy usually leeds to larger plant structures, which is beneficial in breeding programs (larger flowers, differently shaped and sized leaves, etc.). From the perspective of polyploid research, the study reports the opposite case of smaller leaves in new polyploid Rhododendrons, which is interesting.

During review I have identified several shortcommings, which should be explained and improved before acceptance.

I have identified several misreferences in Introduction and Material and Methods. Authors should carefully chack whole text. Overal, the study is poorly referenced.

Flow cytometry was not done in an appropriate way, this part should be better elaborated.

Statictical evaluation was not appropriate.

Style: I suppose that space should be add in front of square bracket containing reference number in the text.

Details

Check:

Table numbering, last table in the article is given as Table 1.

I suggest to find way to merge several tables, when possible.

Introduction

References below do not meet referenced content, authors should carefully check for correct references:

Reference No. 2 – lines 28-29

Reference No. 10 – lines 40-41

There are many classical references which should be used instead No. 3 and 4 in lines 32-35.

Material and Methods

Paragraphs 4.1-4.4 should be better referenced, no reference appeared in these parts of Material and Methods.

Flow cytometry, 4.4 paragraph

Authors should report softwere for flow cytometry evaluation.

The value for coefficient of variation (CV below 0.5) reported for flow cytometry estimations does not make sense. This CV value should be below 5%. It is almost impossible to reach 0.5%.

Line 298 is unnecessary. Data are not further evaluated in Results. And it is not necessary, it is not methodological flow cytometry study.

The authors have adopted apparently the method which is not recommended in flow cytometry community. Flow cytometry was performed without standardisation. However even in such case, rough ploidy level may be determined, but authors should report variation in mean channel position of recorded 2C, 4C or 8C peaks on flow cytometry histograms, not only to report approximate relative fluorescence value as in paragraph 2.2 (in Results). Figure 2 should contain information what a number is indicated in the plot.

Reference No. 33 does not meet content in lines 309-310 (ethanol vs. acetone isolation used in Manzoor et al. 2018)

Authors should use ANOVA instead t test, they must compare usually three groups, diploids, tetraploids and octoploids (4.8 paragraph). Authors should mention precisely what kind of test did they use, if the data were transformed or not and if and how they corrected the test for unequal sample sizes.

Results

I encourage the authors to report mean values and errors whenever possible.

Lines 147-148, content needs to be rephrased

Reference No. 20 does not meet referenced content – lines 228-230

I do not understand how proliferation rate was calculated (2.7 paragraph)

Figure 6D in its present form is not acceptable and even not necessary. Authors were not able to estimate DNA content in absolute units (Mbp) without standardisation (or at least they did not report this in Methods). Moreover one should be surprised that they find precise doubled DNA content when compared 2x vs. 4x and 4x vs. 8x. Error rate of flow cytometry machines is usually 1-3%.

Discussion

Lines 240-241, content needs to be rephrased

In paragraph 240-258, the authors use “multiplication” in strange ways.

I suggest that authors should add missing information mostly to flow cytometry methodology, since they did not use standardisation. Further the article should be better referenced.

Reconsider after major revision.

Reviewer 2 Report

The paper deals within the scope of the journal. Though several papers on Rhododendron induced-polyploidy have been already published, it presents new and original data on the effects of colchicine on R. fortunei's polyploidization.

Line 18: ... epidermal hairs than those of diploids. Furthermore ...

Line 23: please delete "fortunei Lindl." already present in the title.

The Introduction clearly describes the state of the art of the topic as well as the studies previously conducted. This section should be improved by citing recent articles on the same topic as the following one: Manzoor, A., Ahmad, T., Bashir, M. A., Hafiz, I. A., & Silvestri, C. (2019). Studies on colchicine induced chromosome doubling for enhancement of quality traits in ornamental plants. Plants, 8(7), 194.

Line 90: ... darker color with respect to those of the diploids. For tetraploids ...

All the tables are not uniformly formatted and should be necessary presented according to a unique style format.Table 2: in any column statistical differences (letters or standard errors) should be added.Table 5: in the 2nd column please add "Total chlorophyll".

Line 275: please delete "as shown in Table 1", so the new sentence is "Seven colchicine concentrations (0,0.025, 0.05, 0.1, 0.15, 0.2 and 0.25%) and three times (24, 48 and 72h) were set up, thus creating ..."

Line 277: delete "different times according to Table 1", so the sentence is "... solution for 24, 48 and 72h, and then ..."

Table 1 should be removed from the manuscript.

Reviewer 3 Report

COMMENTS TO THE AUTHORS

The manuscript is of scientific interest, however it needs to be improved in some of its parts. It is imperative that origin of plant material and the phases of in vitro cultivation have to be better described. 

Specific comments

Abstract

Line 10: delete (R. fortunei)

Keywords

Line 23: Rhododendron fortunei Lindl. and colchicine are already reported in the title and therefore have to be replaced

Introduction

Line 42: few reported instances … which and what do they talk about, you need to report this information in the introduction

Line 47: this sentence is a conclusion, should be removed

Results

Line 61: seedlings regenerated????   How can seedlings regenerate? Please see comments reported for Lines 281 and 290.

Line 79: This outcome suggests

Line 91: leaf area???? Please, confirm

Line 120 Figure 6????? What does it have to do with it

Tables 4 and 5: please, report all values with 2 decimals, check the spacing and align

Lines 141-149: rewrite in a more correct form

Lines 154-155: this sentence makes no sense

Line 156: multiply the seedlings, what does it means? Seedlings or adventitious shoots?? This is not a correct way to express an in vitro culture. Moreover, you never talk about ‘average proliferation coefficient’ before, how is it calculated?

Lines 159-160: induction rate of callus was never mentioned before in M&M???? Moreover octoploid callus induction???

Lines 161-162: incorrect grammatical form

Discussion

Line 169: has not previously been reported???? in the introduction it is specified that there are reported instances (see line 42)

Line 200: germination of lateral bud in stem bases???? but since when the buds germinate????? producing large amount of callus ??? Please, explain and see comments of Lines 159-160.

Line 233: but are there any support hypotheses? possibly implement

Lines 240-253: this part has to be rewritten in a clearer and more correct form

Materials and methods

Lines 270-273: this part has to be improved and better specified. Moreover, the grammatical form is incorrect.

Clone ZL1, please give its characteristics and origin. We don't know anything about it, why was it chosen? Where do seeds come from?

Seedlings were propagated????? or were germinated ????

Grew ??? means where they grew ???

Stem bases … dimensions?

Other medium information, pH, agar, etc.?????

Line 281: it is assumed that callus has formed and regeneration has occurred, but all this information is missing. If

Lines 281 and 290: the term ‘regenerated seedlings’ does not exist. This is a basic concept to be clarified: vegetal materials regenerated from callus are adventitious shoots that have nothing to do with seedlings. Please clarify!!!!

Line 294: 30 seedlings, randomly choosen?

Conclusion

Line 326: tissue culture seedlings ????????

Lines 329-330: 69 and 29 out of …, please, report

Round 2

Reviewer 1 Report

Rereview of paper „Colchicine-induced Polyploidy in Rhododendron fortunei Lindl.“ by Lan Mo, Junhao Chen, Xiongzhen Lou, Qiangwei Xu, Renhui Dong, Zaikang Tong, Huahong Huang and Erpei Lin

Authors did a good job to improve and respond my comments. However not all of them were addressed satisfactorily.

My concern regards mostly flow cytometry screenings. Authors declared that they use Populus trichocarpa as reference standard. However, the way, how they reported the results in article (Figure 2 and Table 1), indicates that standardisation was external. It means, that they firstly measure P. trichocarpa and then Rhododendron sample as another sample (reference standard and Rhododendron were not prepared and analysed simultaneously in one sample). External standardisation is no longer recommended in flow cytometry screenings. However I do not want to throw away the whole study. But authors should follow recommendations given below:

  1. Since any reports of genomes size based on external standardisation is misleading, I suggest to omit this part from whole study. It is not necessary to report genome size for diploids-octoploids in the present study. Reports on ploidy levels is enough.
  2. Values of third column in Table 1 (Peak value) are clear enough. There is no need to do more. Just clarify number of samples (note given below).
  3. Exclude fourth column in Table 1 (Genome size / Mbp) and also very misleading fourth row of Table 1 with data for P. trichocarpa.

„The test was repeated at least three times for each sample“ (lines 294-295). Authors should here to explain from what number of samples was than calculated mean and standard errors of Peak value in Table 1. If each of samples was repeated three times, then they got 90 values (3 x 30) for diploids, 207 values (3 x 69) for tetraploids and 87 values (3 x 29) for octoploids. Of course authors maybe calculated firstly mean value per sample based on 3 repeats and then averaged for 30, 69 and 29, diploids, tetraploids, octoploids, respectively. This must be clear in methods and in table 1.

Authors concluded that „69 tetraploids and 29 octoploids plants of R. fortunei were obtained from 540 plantlets examined of all treatments“ (lines 328-329). But only 30 diploid plants were analysed for flow cytometry as given in Table 1. This incostinency must be explained. What about rest more than 400 plantlets? How do authors know that they are not polyploids? This is very important, because if 400 plantlets were not screened then any Induction rate calculation is misleading!!! Moreover in line above change to „69 tetraploids and 29 octoploids of R. fortunei were...“

Authors did not report all values in main text as means and error rate, as requested. They referenced to tables, but even in such cases, they should indicate, that given value is „average“ or something similar in this sense.

Authors should add information on sample size in each of tables. We are now not sure, if they measured all 540 plantlets or only that confirmed with flow cytometry or only few of them.

Check form of citation 31 (line 411)
